# Decadal Changes of Organic Carbon, Nitrogen, and Acidity of Austrian Forest Soils

**Robert Jandl * , Ernst Leitgeb and Michael Englisch**

Austrian Forest Research Center, Seckendorff Gudent Weg 8, A-1131 Vienna, Austria;
ernst.leitgeb@bfw.gv.at (E.L.); michael.englisch@bfw.gv.at (M.E.)
* Correspondence: robert.jandl@bfw.gv.at; Tel.: +43-664-826-99-07

**Abstract:** Repeated soil surveys provide opportunities to quantify the effect of long-term environmental change. In recent decades, the topics of forest soil acidification as a consequence of acidic deposition, the enrichment of forest ecosystems with nitrogen, and the loss of carbon due to climate change have been discussed. We used two forest soil surveys that were 20 years apart, in order to establish the direction and magnitude of changes in soil carbon, nitrogen, and soil acidity. Soils have been initially sampled in the late 1980s. The plots were revisited twenty years later. Archived soil samples from the first survey were reanalyzed with the same protocol as the new samples. We found changes in the stocks of soil organic carbon, soil nitrogen, and soil pH. However, the changes were inconsistent. In general, as many sites have gained soil organic carbon, as sites have lost carbon. Most soils have been slightly enriched with nitrogen. The soil pH has not changed significantly. We conclude that changes in the evaluated soil chemical properties are mainly driven by forest management activities and ensuing forest stand dynamics, and atmospheric deposition. We have no convincing evidence that climate change effects have already changed the soil organic carbon stock, irrespective of bedrock type.

**Keywords:** forest soil chemistry; forest soil survey; soil organic carbon; soil nitrogen; soil acidity; Austrian forest soils





## 1. Introduction

Soil chemical properties are remarkably inert and are known to be slow responders to changing site conditions. Despite rapid changes in seasonal climate conditions, changes in the above- and below-ground litterfall density from the herbaceous vegetation, shrubs, trees, and soil organic carbon stocks remain stable over time. Soil pH is buffered due to the interaction of soil water with the soil matrix. The stocks of soil organic carbon and nitrogen are huge, as compared to the annual fluxes of these elements.

The reliable detection of changes of soil chemical properties and the identification of the main drivers of change are challenging. Soil sampling is a destructive process, and a previously sampled spot cannot be re-sampled. Yet, soils are spatially variable. Hence, the 'signal' of temporal change of soil chemical soil properties is overlain by 'noise', due to spatial variability [1,2]. In order to corroborate whether or not a soil chemical property has changed, it is either possible to analyze a large number of replicates, or to re-sample soils after a long time. Based on data from the German Forest Soil Survey, it was expected that a significant enrichment of soils with nitrogen would be detectable after 20 years, whereas the detection of significant changes in the stocks of soil organic carbon would remain elusive [3].

There are both scientific and political reasons for interest in changes of soil properties. Soil pH has received a lot of attention in the context of forest decline in the Northern hemisphere, particularly in the 1980s [4–6]. With respect to soil acidification, the main issue is an irreversible soil detoriation, mostly due to the destruction of clay minerals at very

acidic sites. Soil properties can be affected long term by adverse impacts on the nutrient and water retention capacity.

With respect to soil nitrogen, a major concern was the induction of imbalanced tree nutrition, due to elevated rates of the deposition of atmospheric compounds containing nitrogen. When plants are growing according to an increasing supply of nitrogen, the supply of other nutrients may not keep up, and nutrient deficiencies may develop [7]. The topic of nitrogen eutrophication and nitrogen saturation triggered major research efforts, with quite controversial opinions about the benefits and problems on nitrogen enrichment, depending on the metrics that have been used. Many forests have responded to elevated nitrogen deposition with higher growth rates, yet nitrate leaching from soils to aquifers and a decline is plant species richness remained a concern [8–12]. The debate on the nitrogen enrichment of forests due to atmospheric deposition is quite vivid, and continues in the discussion on biodiversity losses.

The most recent soil-related topic is the impact of climate change on soil organic carbon. The sheer size of the soil organic carbon stock raises the question as to whether the stock is stable over time [13]. Two main lines of argument are brought forward. Firstly, increases in tree productivity due to global warming increase the rate of biomass production. More above-ground and below-ground litterfall can increase the soil organic carbon stock. On the contrary, increased temperatures are stimulating soil decomposition processes and lead to a decrease in soil organic carbon stocks. An important question for managed forest ecosystems is whether forest management strategies or land management in general can possibly affect the direction of changes in soil organic carbon stocks [14,15], thereby incorporating forest soils in climate change mitigation strategies. A well known effort is the 4-per-mil concept that implies that minor changes in the soil organic carbon stock may have a relevant impact on climate change mitigation [16,17]. Yet, numerous socio-economic and political impediments may severly limit achievable soil carbon sequestration, thereby reducing the role of soils for climate change mitigation [18].

The mechanisms of soil organic carbon storage in soils are well investigated [19–22]. Yet, it is still elusive to define a reference level for soil organic carbon stock for forests under given site and management characteristics [23,24]. Climatic parameters, soil texture, and soil oxides are identified as valuable predictors for agricultural soils [25,26]. The biogeochemical cycle of nitrogen is tightly linked to the cycle of organic carbon. Resolving soil organic carbon dynamics is an integral part of understanding soil nitrogen and involves both anthropogenic and natural drivers of ecosystem dynamics.

Recently, the stocks of soil organic carbon in Austria have been assessed as part of the CarboSeq project of FAO [27]. The study included all types of land use. It was confined to the upper 30 cm of the soil and focussed on carbon in order to maximize the number of participating countries. The Austrian contribution is explained in detail in a separate publication. Accordingly, Austrian forest soils hold 128 t C/ha in the organic surface layer and the upper 30 cm of the mineral soil [28].

In this paper, we use the available data of the Austrian Forest Soil Survey and the BioSoil project in order to identify changes in soil acidity and soil organic carbon and nitrogen [29,30]. We evaluate the data from a repeated forest soil inventory and include the organic surface layer and the mineral soil to a depth of 50 cm. We hypothesize that soil nitrogen has increased significantly within the 20 years of our study, whereas changes in soil pH are small. The hypothesis is supported by an analysis of atmospheric deposition trends. Already before the first soil survey in 1989, effective measures had been adopted to reduce the emissions of sulphur dioxide, which was a main cause for soil acidification. Yet, the emissions of nitrogen oxides remained at high levels [31–33]. We further hypothesize that eventual changes in soil organic carbon will not be statistically significant. The expected decline due to higher decomposition rates of soil organic matter is partially compensated by increased carbon inputs to the soils due to increased forest productivity. In addition, numerous measured and unmeasured parameters contribute to the considerable spatial

heterogeneity of soil organic carbon stocks. This hypothesis is supported by experiences from several other national forest soil inventories [29,34–36].

## 2. Materials and Methods

### 2.1. Sites

The sites of the soil investigation are located on the regular grid of the Austrian Forest Inventory [37]. On more than 500 sites, the initial forest soil survey has taken place, and soil analysis was finished in 1989 [30]. Twenty years after the initial assessment, a repetition was made within the BioSoil project (http://icp-forests.net, accessed on 15 February 2022). Within the BioSoil, only 139 sites were selected, and soil samples for chemical analysis were collected. Rather than sampling soils from pedogenetic soil horizons, in both surveys, samples were taken from fixed depth steps of the mineral soil. The sampling protocol has been changed between the surveys in order to support a European harmonization effort. Whereas the Austrian Forest Soil Survey used the separation of 0–10, 10–20, 20–30, 30–50, and 50–80 cm, the BioSoil survey used 0–5, 5–10, 10–20, 20–40, and 40–80 cm. At each sampling point, four soil pits were opened, and the collected samples were pooled, in order to obtain one representative sample per soil horizon and plot. The rock content of the soil horizons was visually estimated in the field. The organic surface layer was sampled separately. A quadratic steel frame with 30 cm side length was put on the surface, and the organic material inside the frame was collected. Samples were dried and weighed, and chemically analyzed with the same protocol as samples of the mineral soil.

The site characteristic for subsetting the dataset in our analysis is bedrock, distinguishing between soils derived from calcareous or silicatic bedrock. Pragmatically, every soil profile where carbonate was detected in the field test (fizzing when applying diluted HCl) was grouped to 'calcareous soils', otherwise to the group of 'silicatic soils'. This dichotomy was chosen because the geological characteristics, as shown in a geological map, incompletely reflect pedological site conditions at some sites. Examples are sites in Upper Austria where the geological map shows silicatic schists. Yet, in some places, the bedrock is overlain by fluvially transported calcareous quarternary material.

The climatic characteriziation of the sites is available by the mean annual temperature and precipitation for the period 1960–90. The climate data were provided by the Zentralanstalt für Meteorologie und Geodynamik (ZAMG; http://www.zamg.ac.at, accessed on 15 February 2022) in Vienna. The climate data for the sampling sites were interpolated from the network of climate stations. An elevation–correction of measured data was necessary, as mountain regions are insufficiently represented by climate stations [38].

### 2.2. Soil Analysis

Soil samples of both soil surveys were delivered to the lab of the Austrian Forest Research Center, and were air dried. A part of the samples was stored in a soil archive in order to allow later re-analysis. The samples were analyzed according to the ICP manual [39]. Briefly, concentrations of organic carbon and nitrogen were analyzed with a Carlo-Erba combustion analyzer. Soil pH was determined in a 0.01 m $CaCl_2$ slurry. Data integrity was ensured by benchmarking the applied measurement protocols within the international interlaboratory comparison that was organized by ICP-Forests [40]. The laboratory protocols were updated whenever its data deviated from benchmark values of the round-robin tests. A potential bias of old and new soil chemical data is caused by changes in the technical infrastructure of the laboratory, and due to changing laboratory staff. In order to avoid this bias, archived soil samples were re-analyzed. For each of the plots of the BioSoil project, the soil samples were analyzed together with retrieved soil samples from the initial survey. An unpublished comparison of the data showed that the concentrations of soil organic carbon and nitrogen and the $pH_{CaCl_2}$ have not changed in the air-dried archived samples during 20 years of storage. Therefore, we are confident that eventually, detected differences in soil chemical properties will reflect soil changes and will not be consequences of a bias due to inconsistencies in sample stability and laboratory protocols.

Soil texture was assessed with the pipette method ([39] Part X, Method SA03). In the soil survey of the year 1989, soil texture at each site was measured for the deepest sampled horizon of the mineral soil. A preliminary unpublished project has shown that the deepest layer of the mineral soil is representative of the entire soil profile. The obtained particle size distribution was taken as a stable site property, and was used for the calculation of organic carbon and nitrogen stocks for both soil surveys.

Soil bulk density $\rho$ was estimated with a function that has been derived from a database of Austrian forest soils. The predictors of $\rho$ are the concentration of soil organic carbon and soil texture classes [41].

In order to account for different soil horizon depths that were used in the sampling protocols of the two soil surveys, we split each horizon in 1 cm-slices and assigned to each slice the soil chemical and physical properties of the respective soil horizon. With rock content, *eho*, and the depth of soil horizons the mass of fine soil, i.e., particles < 2 mm, per area (kg fine soil/$m^2$/cm) was calculated. Fine soil mass was multiplied with the concentrations of organic carbon and nitrogen, in order to obtain the masses of organic carbon and nitrogen. The stocks of organic carbon and nitrogen in the mineral soil are represented by the cumulated values of the 1 cm-slices. The stocks of organic carbon and nitrogen in the organic surface layer were obtained from the multiplication of the mass of the organic layer and the respective concentrations of carbon and nitrogen. Annual changes of the stocks of organic carbon and nitrogen for each site were calculated as differences between the respective stocks in the two surveys, divided by the time between surveys, i.e., 20 years.

Neither the initial forest soil survey of 1989 nor the BioSoil project of 2009 included soil ecological parameters.

*2.3. Data Evaluation*

Soil data were statistically evaluated. We confined our analysis to the organic surface layer and the upper 50 cm of the mineral soil. The number of sites where the mineral soil extend below 50 cm was small in both soil surveys and 50 cm was a reasonable cut-off.

A comparison of the concentrations and stocks of organic carbon and nitrogen, and the pH-value, respectively, in both soil surveys was made by pairwise *t*-tests. The data were than stratified further according to 'soil survey' and 'geological bedrock'. Differences between strata were analyzed by ANOVA and a subsequent multiple comparison of means (Tukey-HSD test). For data processing, statistical analysis, and graphics, we used R v.4.1.1 ('Kick Things') and the packages AQP, agricolae, MASS, lattice, dplyr, and ggplot [42–47]. Our analysis uses the subset of data, where both initial and repeated chemical soil analyses are available, and where the required ancillary data are available. For different parameters, the sample size therefore varies slightly.

We were interested as to whether our data reveal convincing predictors of soil organic carbon sequestration. Mean annual air temperature and soil texture were, among others, proposed as candidate predictors [26]. We calculated correlations between individual sites (R functions pairs and cor.test) and a best-fit model was obtained with a stepwise, multiple forward regression in order to scrutinize the predictors with data that are available from our sites (R package MASS, functions lm and stepAIC) [45].

**3. Results**

The differences in the concentrations of organic carbon, nitrogen, and pH in soil that have developed over the course of 20 years are shown in Table 1. The comparison is confined to the upper 30 cm of the soil, where comparable sampling depths have been used in both surveys. The organic carbon and nitrogen concentrations are significantly higher in the survey of 2009, whereas soil pH has not significantly changed. In the upper 10 cm of the mineral soil, a highly significant increase in the concentrations of organic carbon and nitrogen took place. The HSD-Tukey test indicated differences between sites on calcareous and silicatic bedrock, respectively. In general, the comparison of means distinguished only

a few groups, due to the high variability within strata. The difference between bedrock types prevails in the entire soil profile, whereas differences in concentrations of organic carbon and nitrogen between the two soil surveys are not significant in the depths 10–20, and 20–30 cm, respectively. The pH-data show consistently the expected trend of higher values in deeper horizons of the mineral soil. The entire data set shows the statistically significant separation between sites on silicatic vs. calcareous bedrock, respecitvely. No temporal trend was detected.

**Table 1.** Comparison of concentrations of carbon and nitrogen (mg g$^{-1}$) and the pH value in a repeated soil survey. The values are the arithmetic mean ($\bar{x}$) and the standard deviation ($sd$) of a sample size of $n$ = 119 sites, with 90 sites on silicatic and 29 sites on calcareous bedrock. Statistical differences between the two soil surveys are given by the $p$-values; differences between the strata 'survey' and 'bedrock' are shown by letters that indicate the grouping according to a comparison of means (Tukey test).

| | Survey 1989 | Survey 2009 | Survey 1989 | Survey 2009 | Survey 1989 | Survey 2009 |
|---|---|---|---|---|---|---|
| | **All Data** | | **Silicatic Bedrock** | | **Calcareous Bedrock** | |
| **Carbon** | | | | | | |
| Forest floor | 330.1 ± 70.4 | 450.9 ± 80.9 | 320.1 ± 70.4 | 442.7 ± 88.2 | 353.4 ± 65.4 | 469.9 ± 58.8 |
| | <0.001 | | bc | a | b | a |
| 0–10 cm | 79.1 ± 65.5 | 103.8 ± 72.5 | 56.8 ± 33.3 | 82.3 ± 50.5 | 130.4 ± 89.3 | 155.4 ± 89.9 |
| | <0.001 | | c | b | a | a |
| 10–20 cm | 42.8 ± 41.2 | 47.2 ± 50.8 | 29.1 ± 21.3 | 31.4 ± 23.0 | 76.8 ± 56.8 | 86.6 ± 75.1 |
| | n.s. | | b | b | a | a |
| 20–30 cm | 28.9 ± 29.1 | 30.6 ± 33.1 | 20.8 ± 17.0 | 20.1 ± 17.5 | 41.9 ± 35.6 | 58.6 ± 46.7 |
| | n.s. | | b | b | a | a |
| **Nitrogen** | | | | | | |
| Forest floor | 12.1 ± 2.2 | 14.3 ± 2.8 | 11.8 ± 2.1 | 13.9 ± 2.7 | 12.9 ± 2.1 | 15.1 ± 2.9 |
| | <0.001 | | c | b | b | a |
| 0–10 cm | 4.0 ± 3.0 | 5.6 ± 3.6 | 2.8 ± 1.5 | 4.4 ± 2.7 | 6.7 ± 1.5 | 8.5 ± 4.1 |
| | <0.001 | | d | c | b | a |
| 10–20 cm | 2.3 ± 2.1 | 2.7 ± 2.7 | 1.5 ± 0.9 | 1.7 ± 1.2 | 4.5 ± 0.9 | 5.1 ± 3.6 |
| | n.s. | | b | b | a | a |
| 20–30 cm | 1.6 ± 1.4 | 1.8 ± 2.0 | 1.1 ± 0.7 | 1.1 ± 0.8 | 2.9 ± 1.9 | 3.6 ± 2.8 |
| | n.s. | | b | b | a | a |
| $pH_{CaCl_2}$ | | | | | | |
| Forest floor | 4.2 ± 0.9 | 4.3 ± 1.0 | 3.9 ± 0.7 | 3.9 ± 0.7 | 5.1 ± 0.8 | 5.3 ± 0.8 |
| | n.s. | | b | b | a | a |
| 0–10 cm | 4.6 ± 1.4 | 4.5 ± 1.3 | 3.8 ± 0.5 | 3.7 ± 0.5 | 6.4 ± 1.0 | 6.2 ± 1.1 |
| | n.s. | | b | b | a | a |
| 10–20 cm | 4.9 ± 1.3 | 4.9 ± 1.3 | 4.2 ± 0.5 | 4.1 ± 0.4 | 7.2 ± 0.6 | 6.6 ± 1.0 |
| | n.s. | | b | b | a | a |
| 20–30 cm | 5.0 ± 1.3 | 5.1 ± 1.3 | 4.4 ± 0.5 | 4.3 ± 0.5 | 7.0 ± 0.8 | 6.9 ± 0.9 |
| | n.s. | | b | b | a | a |

Small differences have led to slightly higher stocks of soil organic carbon and nitrogen in the forest floor and the upper 50 cm of the mineral soil over the course of 20 years. The differences are highly significant for the forest floor, and statistically insignificant for the mineral soil. Within the surveys, significant differences between sites with calcareous vs. silicatic bedrock were identified (Table 2). The accumulation of organic carbon and nitrogen is larger on the forest floor than in the mineral soil. Visual inspection suggests that, at most sites, the changes are smaller than the range of the standard deviation around the mean. Yet, a shift towards higher stocks of organic carbon and nitrogen took place within 20 years between the surveys. Silicatic and calcareous soils have changed over time in a similar way, as indicated by the proximity of the linear regression functions (Figure 1).

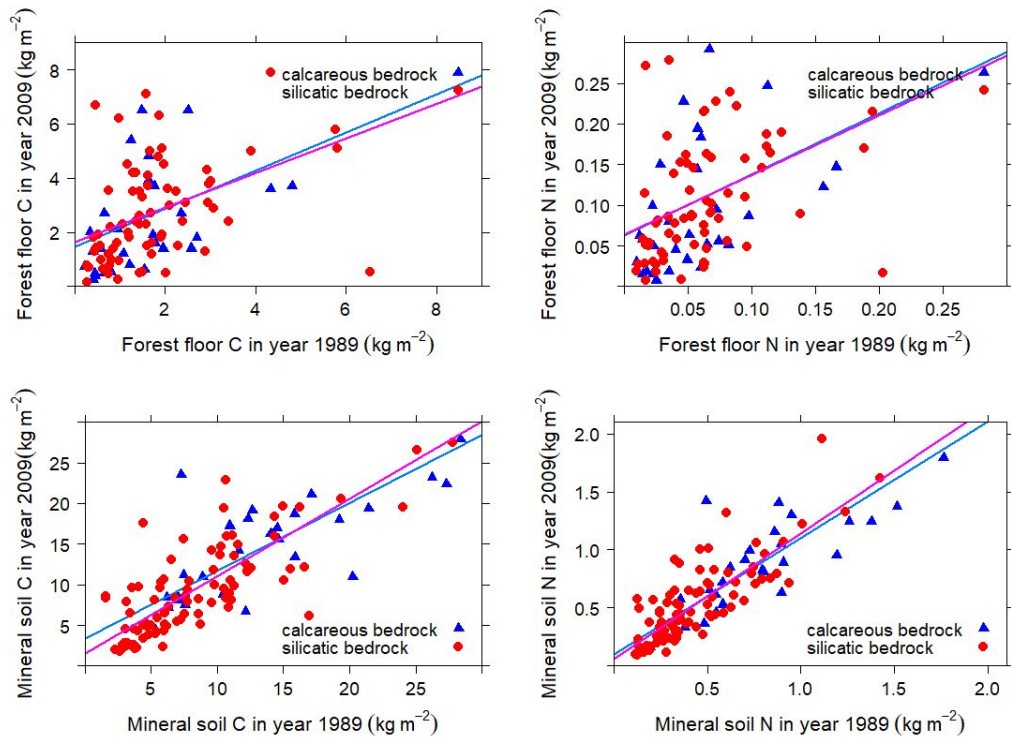

**Figure 1.** Change of stocks of soil organic carbon and nitrogen in Austrian forest soils within 20 years. Upper panel: Change in the litter layer. Lower panel: Change in the mineral soil to a depth of 50 cm (**left**: organic carbon, **right**: nitrogen). The blue triangles show sites on calcareous bedrock, the red circles sites on silicatic bedrock, respectively. The lines indicate the linear regression functions.

**Table 2.** Stocks of soil organic carbon and nitrogen ($kg\,m^{-2}$) in the organic surface layer and the upper 50 cm of the mineral soil. Statistical differences between the two soil surveys are given by the *p*-values; differences between the strata 'survey' and 'bedrock' are shown by letters that indicate the grouping according to a comparison of means (Tukey test).

| | Survey 1989 | Survey 2009 | Survey 1989 | Survey 2009 | Survey 1989 | Survey 2009 |
| --- | --- | --- | --- | --- | --- | --- |
| | **All Data** | | **Silicatic Bedrock** | | **Calcareous Bedrock** | |
| Carbon | | | | | | |
| Forest floor | $1.8 \pm 1.4$ | $2.7 \pm 2.0$ | $1.9 \pm 1.5$ | $2.7 \pm 1.9$ | $1.5 \pm 1.1$ | $2.5 \pm 2.3$ |
| | | <0.001 | b | a | b | ab |
| Mineral soil | $10.0 \pm 7.4$ | $11.1 \pm 7.8$ | $8.5 \pm 5.4$ | $9.7 \pm 6.5$ | $14.7 \pm 10.3$ | $15.6 \pm 9.6$ |
| | | n.s. | b | b | a | a |
| Nitrogen | | | | | | |
| Forest floor | $0.06 \pm 0.05$ | $0.11 \pm 0.09$ | $0.07 \pm 0.05$ | $0.11 \pm 0.08$ | $0.05 \pm 0.04$ | $0.10 \pm 0.10$ |
| | | <0.001 | bc | a | bc | ab |
| Mineral soil | $0.53 \pm 0.41$ | $0.63 \pm 0.47$ | $0.43 \pm 0.26$ | $0.53 \pm 0.35$ | $0.85 \pm 0.59$ | $0.95 \pm 0.64$ |
| | | n.s. | b | b | a | a |

An evaluation of individual sites, stratified by bedrock material (soils derived from silicatic vs. calcareous bedrock), is shown in Figure 2. The sites were ordered according to the size of the change on stocks of soil organic carbon and nitrogen over time. The figure shows very few sites with surprisingly large changes. Most sites have almost negligible annual changes of the stocks of organic carbon and nitrogen, indicating that soils have overall been enriched with organic carbon and nitrogen. Very large gains and losses of either element cannot be explained with ecological processes, and are possibly artefacts due to peculiar local conditions that are not reflected in the data. Yet, we had no evidence for measurement errors or other sources of bias.

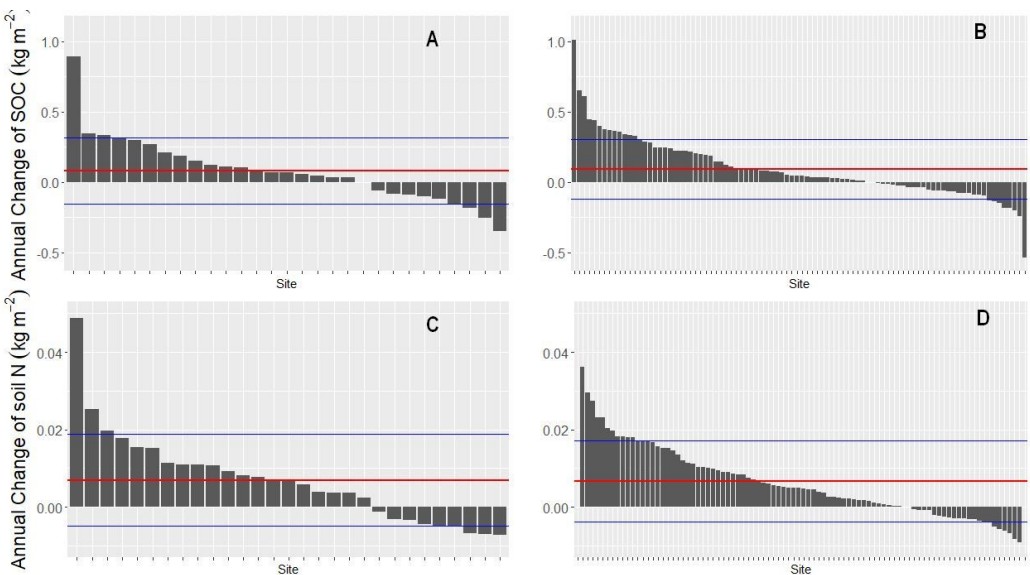

**Figure 2.** Annual change of organic carbon and nitrogen stocks in Austrian forest soils on 139 sampled sites ordered according to size of the change in the respective stocks. The thick horizontal red line shows the mean annual change, the fine horizontal blue lines show the mean ± the standard deviation. The annual change of organic carbon is shown in (**A**) for sites on calcareous bedrock and (**B**) for sites on silicatic bedrock. The annual change of nitrogen is shown in (**C**) for sites on calcareous bedrock and (**D**) for sites on silicatic bedrock.

The extent of soil acidfication is shown in Table 1 and in Figure 3. There is no indication that forest soils have acidified within the 20 years between the soil surveys. A de-acidification is not evident either. The range between the 25- and 75 percentiles is narrower for soils derived from silicatic bedrock. The wider range for calcareous soils can is explained by soils that are superficially acidified, but carbonate is still present in the subsoil. Such soils are encountered in Upper Austria, where quarternary deposits have been accumulated after the last glaciation and that have acidified since then. The forest soils are often acidic in the organic surface layer and the upper mineral soil.

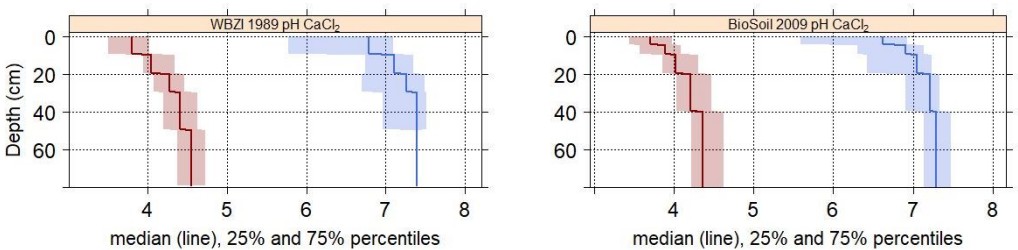

**Figure 3.** Extent of change in soil acidity over 20 years. Depth gradient of the median and the 25 and 75 percentiles of the pH in the mineral soil. Red lines and shades represent sites on silicatic bedrock, blue lines and shades represent sites on calcareous bedrock. **Left** graph: Forest soil survey in year 1989; **right** graph: BioSoil survey in year 2009.

Soil process understanding tells that climatic factors and mineralogical properties are indicative of the long-term carbon storage in soils [26]. Of major relevance is the content of clay minerals and silty materials that provide coupling sites for organic molecules in the mineral soil. Such considerations are relevant for the quantification of the carbon sequestration potential in soils. An obvious factor for the assessment of soil organic carbon stock changes is air temperature, because warmer sites allow for higher soil microbial activities and may trigger a depletion soil organic carbon stocks. In Figure 4, we investigate whether soil organic carbon, both expressed as concentration and stock, is related to the

clay or clay-plus-silt content, or to the annual mean air temperature, respectively, at our investigated sites. Figure 4 does not indicate a strong correlation between the independent variables soil texture and air temperature, and soil organic carbon as dependent variable. The correlation coefficients and their respective statistical significances are shown in Table 3. A multiple stepwise regression with the concentration of organic C in the upper 30 cm of the mineral soil as dependent variable and the annual mean air temperature, and percentage of clay and silt, respectively, as independent variables entered temperature and silt and yielded a modest correlation ($R^2 = 0.12$). The respective correlation for the organic carbon stock in the upper 30 cm of the mineral soil had an $R^2$ of 0.10. The relevance of air temperature for the prediction of organic carbon was much higher than the relevance of soil texture. Overall, mean annual air temperature and soil texture are predictors of the concentration and stock soil organic carbon with only moderate predictive power (Table 4).

**Table 3.** Correlation between soil organic carbon concentrations (mean carbon concentration of upper 30 cm of the mineral soil) and soil organic carbon stocks (sum of organic carbon in upper 30 cm of the mineral soil) with the clay content (%), the sum of clay and silt (%), and the mean annual temperature ($\overline{T}$ (°C)), average of the years 1960 to 1990), respectively. The analysis is based on data from 429 sites of the Austrian Forest Soil Survey 1989.

|  | **Concentration of Organic Carbon** | | **Stock of Organic Carbon** | |
|---|---|---|---|---|
|  | **Correlation Coefficient** | ***p*-Value** | **Correlation Coefficient** | ***p*-Value** |
| clay | −0.03 | 0.47 | −0.06 | 0.22 |
| $\sum$(silt + clay) | 0.08 | 0.10 | 0.05 | 0.26 |
| $\overline{T}$ | −0.30 | <0.001 | −0.28 | <0.001 |

**Table 4.** Equations for estimating the soil organic carbon concentration (mg C/g) and the soil organic carbon stock (kg C/m$^2$), respectively, from mean annual temperature (°C) and the sum of silt and clay (%). The coefficients have been derived from a multiple stepwise regression. 'DF' ...degrees of freedom, R$^2$ ...coefficient of determination, RSE ...relative standard error. The statistical significance of the intercepts and coefficients is given by ⋆⋆⋆ ...$p < 0.001$, ⋆ ...$p < 0.05$ and 'n.s.' ...'not significant'. The analysis is based on data of the Austrian Forest Soil Survey of 1989.

| **Sites** | **Intercept** | **Temperature** | **Silt + Clay(%)** | **DF** | **R$^2$** | **RSE** |
|---|---|---|---|---|---|---|
| Soil organic carbon concentration | | | | | | |
| all sites | 100.7 ± 15.6 | −7.1 ± 1.0 | −0.09 ± 0.2 | 475 | 0.09 | 37.7 |
|  | ⋆⋆⋆ | ⋆⋆⋆ | n.s. | | | |
| silicatic sites | 115.6 ± 16.8 | −8.5 ± 0.9 | −0.3 ± 0.2 | 340 | 0.22 | 27.5 |
|  | ⋆⋆⋆ | ⋆⋆⋆ | n.s. | | | |
| calcareous sites | 48.0 ± 29.4 | −4.9 ± 2.7 | 0.7 ± 0.3 | 132 | 0.07 | 49.3 |
|  | n.s. | n.s. | ⋆ | | | |
| Soil organic carbon stock | | | | | | |
| all sites | 12.4 ± 1.9 | −0.7 ± 0.1 | | 473 | 0.07 | 4.6 |
|  | ⋆⋆⋆ | ⋆⋆⋆ | | | | |
| silicatic sites | 14.9 ± 2.4 | −0.9 ± 0.1 | | 339 | 0.13 | 3.9 |
|  | ⋆⋆⋆ | ⋆⋆⋆ | | | | |
| calcareous sites | 7.0 ± 3.3 | −0.5 ± 0.3 | 0.1 ± 0.0 | 131 | 0.06 | 5.5 |
|  | ⋆ | n.s. | ⋆ | | | |

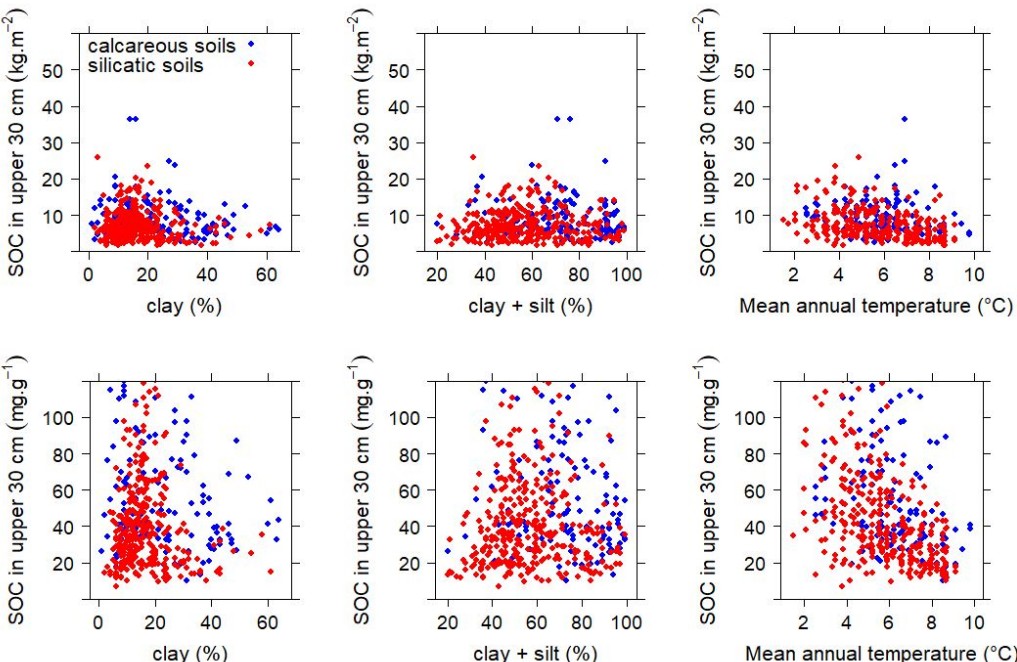

**Figure 4.** Relation of soil organic carbon stock, and concentration, respectively, soil texture and air temperature in Austrian forest soils. Upper panel: Relation of organic carbon stocks with the clay content (**left**), the sum of clay and silt (**center**), and air temperature (**right**). Lower panel: Relation of organic carbon concentrations with the clay content (**left**), the sum of clay and silt (**center**), and air temperature (**right**). Air temperature represents the mean of the years 1960–1990. The different color represent the bedrock material (blue: soils derived from calcareous bedrock; red: soils derived from silicatic bedrock.

## 4. Discussion

Soils are an integral compartment of biogeochemistry and are reflecting environmental change. Soils receive element input from the atmosphere, the vegetation, and from geochemical processes such as rock weathering. Necessarily, changes in the rates of biogeochemical processes eventually affect soil chemical properties [19,48]. Yet, changes in chemical soil processes are often slow. In particular, changes of large stocks such as soil organic carbon and nitrogen are responding slowly to external processes except for extreme events such as massive erosion or accumulation of soil material. With many anthropogenic influences on soils, the detection of soil changes receives increasing attention [49].

The availability of large-scale forest soil surveys in many countries, particularly in Europe, has fuelled the question whether anthropogenic environmental change is already evident from the analysis of soil chemical properties. An obvious approach is the repetition of soil surveys. Results of changes in soil chemical properties obtained from field data are required to corroborate the results of simulation experiments. Yet, the high spatial variability makes the distinction of signal and noise difficult. Examples of the successful identification of changes in soil organic carbon, nitrogen, and pH are available from soil monitoring projects in the United Kingom, Switzerland, Germany, and Sweden [50–53]. Evidence from ground truthing is required when anthropogenically induced changes of soil properties are addressed in conceptual studies.

We investigated the change in soil pH, soil organic carbon, and soil nitrogen. Soil pH was under scrutiny when combustion processes have enriched the atmosphere with compounds that acidified forest soils. A successful emission reduction for sulfuric compounds was implemented already in the 1990ies and sulfur dioxide emissions in Austria were greatly reduced within a short time [33]. The shared awareness for potential forest detoriation among European countries has enabled the implementation of efforts towards

massive reductions of $SO_2$ emissions and the load of acidity was greatly reduced. Yet, forest soil acidification was by no mean a new phenomenon of the late 20th century. Biomass harvest for the benefit of agricultural production and bioenergy for a growing human population and evolving cottage industries that led to a slow, yet uni-directional degradation of forest soils [54–56]. Historical changes in agricultural practices and the use of fossil fuels instead of bioenergy alleviated the pressure on forest ecosystems. The question remains, in which time span forest soils can recover from these earlier effects. After all, the only natural de-acidification process is chemical rock weathering [57], which works at a rather slow rate.

In the time span between our two soil surveys no major anthropogenic large-scale soil acidification processes were at work. On the contrary, acidic emissions were reduced, and presumably soil-acidifying tree species such as Norway spruce are gradually replaced by forests that are dominated by deciduous tree species. Rather than expecting further soil acidification, a de-acidification was deemed possible. However, the data do not indicate such a tendency, yet (Table 1 and Figure 3). Table 1 shows that soil pH values are almost identical in both surveys. The pH values show the expected depth gradient and difference between soils derived from calcareous and silicatic bedrock, respectively. No detectable soil acidification or de-acidification has taken place in the last 20 years. The natural recovery of acidified soils is obviously a slow process that could be accelerated by liming at sites where soil acidification is deemed critical. Evidence is given by a large-scale liming campaign in SW-Germany [58,59]. Yet, some of the effects of liming may be transient and further long-term research for its full evaluation is warranted [10,60].

Research on soil organic carbon was process-based for a long time [21,22,61,62]. The consideration that soils represent a huge organic carbon stock and increasing the stock by a minute fraction may be part of successful climate-change mitigation, because capturing and sequestering carbon dioxide from the atmosphere brought the research on soil organic carbon to the forefront of attention [63,64]. Greenhouse gas emission inventories on the national, the European, and the global scale have shown that forest ecosystems act as a sink for carbon dioxide [13,65–67]. However, the role of forest soils is not entirely clear. Where the productivity of the forest is increased due to climate change or where an increase of the forest area takes place, a temporary $CO_2$ sink can be expected. At sites where warming accelerates the decomposition of soil organic carbon a $CO_2$ source is likely [68]. In cases where forest soils are $CO_2$ sinks, the pressure on the implementation of technological reductions of green-house gas emissions is alleviated. Carbon sequestration soils are by no means the ultimate solution, but it can still buy time for the development of technical solutions. In the Green Deal of the European Union soil organic carbon is a recognized key element of climate change mitigation [69]. Programmatic approaches such as the 4-per-mil-initiative are married with conceptual approaches on the soil organic carbon sequestration potential, and are also part of the widely used Roth-C simulation programme [17,26,70]. Yet, some expectations on the role of forest soils towards carbon sequestration are overly optimistic and may serve as a reason to further delay action on sustainable climate change mitigation.

In our analysis we found that even 20 years (approximately a fifth of the rotation period of an average Austrian forest) the changes in soil chemical properties were small and partially statistically insignificant (Table 2, Figures 1 and 2). This finding corroborates the concept that elevated input of organic carbon, not necessarily increasing the soil stock size of organic carbon, to a large extent. Instead, an ecological theory shows that biogeochemical cycles are rather accelerated. A higher availability of soil organic matter due to litterfall increases the decomposition rate of organic compounds. The net effect on the soil organic stock is small [71]. The organic soil layer is rather enriched in organic carbon (Tables 1 and 2). Organic material that is not chemically bound to mineral substances is probably more easily decomposed than organic material that is associated with minerals [62]. The organic carbon stocks in the mineral soil, that hold approximately 80% of soil organic carbon, were affected to a lesser degree (Figure 1). Figure 2 shows,

on average, very small positive and negative annual changes of the carbon stocks. Some large differences cannot be explained on the basis of the data analysis, but could depend on personal biases by the field crew or forest stand dynamics that are not captured in the recorded data. The unclear pattern indicates that soil organic stock changes are influenced by several factors, e.g., numerous effects of forest management, that may have a stronger immediate effect on soil organic carbon stocks than climate change.

We had hoped to identify a metric for the carbon sequestration potential of Austrian forest soils. Such a benchmark would be instrumental to constrain the expectations on the potential and technically achievable contribution of forest soil organic carbon in climate change mitigation. In case studies, climatic factors and soil texture have been identified as good predictors of the carbon sequestration potential [26]. These parameters are also key in the widely used RothC model [72]. In Figure 4 we show that neither concentrations nor stocks of organic carbon are correlated with soil texture nor mean air temperature (Table 4). An upper limit for the expectable increase in soil organic carbon, i.e., a benchmark for potential carbon stocks, is not available and cannot be derived from the available data of the two soil surveys.

The biogeochemical fluxes of carbon and nitrogen are closely interlinked. In our survey, we found a clear signal towards increased nitrogen stocks. Again, the signal is stronger in the forest floor material than in the mineral soil (Tables 1 and 2, and Figure 1). Obviously, deposited nitrogen was withheld in the soil effectively. This is a consequence of the prevailing nitrogen limitation in Austria's forests. Despite decades of high nitrogen deposition the essential nutrient is still effectively retained. Centuries of exploitative forest use have reduced the nitrogen stocks that are still not fully replenished [73]. Therefore, nitrogen is incorporated in the biogeochemical cycles, and is readily re-absorbed by plants once it is released by the decomposition of soil organic matter. Nitrogen eutrophication as a threat to forest ecosystems is still discussed. However, the threat of ground water pollution due to nitrate leaching is only locally an issue, whereas biodiversity issues are more critical [8,74].

An emerging important aspect of soil monitoring is the assessment of soil ecological parameters. Microbial activity and the community structure of soil organisms are essential for understanding soil processes. The field of soil microbial ecology is quickly evolving and has yielded new insights in soil functioning [75–77]. However, at the time of planning, the forest soil surveys the confidence in robust and expeditive methods for soil biological parameters was low and the surveys were confined to soil physical and soil chemical parameters and site characteristics.

**Author Contributions:** R.J. analyzed the raw data and wrote the paper. M.E. conceptualized the initial soil survey. E.L. coordinated the BIOSOIL project. All authors have read and agreed to the published version of the manuscript.

**Funding:** The project was funded by the Austrian Government and Forest Focus. It is a contribution to the ACRP project CASAS.

**Institutional Review Board Statement:** The study did not involve humans or animals.

**Informed Consent Statement:** Not applicable for studies not involving humans.

**Data Availability Statement:** Data can be requested from the author. The requests require scrutinization against the data sharing policy of the Austrian Forest Research Center.

**Acknowledgments:** We are grateful for the commitment of the field and lab personnel of the Austrian Forest Research Center and to Ambros Berger and Georg Kindermann for data crunching advice. We also acknowledge the contributions of Franz Mutsch and Robert Hacker, who had passed away too early. We also thank the reviewers for highly constructive comments.

**Conflicts of Interest:** The authors declare no conflict of interest.

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
