# Peer review of "Decadal Changes of Organic Carbon, Nitrogen, and Acidity of Austrian Forest Soils"

_soilsystems, doi:10.3390/soilsystems6010028_

Round 1

Reviewer 1 Report

There have been  found only small amendments in spelling and maybe interpunction.

Author Response

Thank you for detecting our spelling errors. We have corrected them. Otherwise, there was support for our manuscript.

The second reviewer has mostly recommended improving figure 1. We have done that. We have changed the captions of figure 1 and 2 accordingly.

Reviewer 2 Report

I don't have any comments. In my opinion, the presented data might be of interest to scientists and foresters. The manuscript is written transparently and explains all the issues related to it. Overall, I  believe that the manuscript can be printed in present form. 

Author Response

Reply to the reviewers

Thank you for the helpful comments.
One reviewer was already satisfied and suggested some spell checking, which was done.

The second reviewer recommended redrawing our figure 1 and to add regression lines. We have done that and have updated the figure caption accordingly. We also slightly amended the caption of Figure 2.

Some minor corrections have been made in the bibliography. No additional references were added.

This manuscript is a resubmission of an earlier submission. The following is a list of the peer review reports and author responses from that submission.

Round 1

Reviewer 1 Report

attached

Author Response

pls see attachment

Reviewer 2 Report

This is an interesting study and is a source of additional knowledge about changes in soil resulting from human activities. This type of research is especially important in connection with the ongoing climate change on Earth. Therefore, I believe that the Authors should emphasize the purpose of the research more. Perhaps it would be worth trying to formulate research hypotheses, especially since the fragment ‘…Biogeochemical processes that introduce alkalinity to the soils are either the deposition of alkaline   dust, chemical rock weathering, or non-acidic litterfall material from the vegetation. These three processes are either very slow or supply small amounts of alkalinity...’could be treated as kind of their verification.

It is a pity that the Authors have omitted in the discussion the importance of soil organisms, even if they play only an indirect role in soil processes. Nevertheless, the lack of information did not detract from the value of this research.

Author Response

pls see attachmetbb

Reviewer 3 Report

The soilsystems-1344159 entitled “Decadal changes of chemical properties of Austrian forest soils” by Jandl et al. reports on results of changes in soil acidity and soil organic carbon and nitrogen between 1989 and 20 years later. In major detail, the research analyzed soil data from 119 sites, and compared the data from 1989 and 20 years later. It has a lot of work and a certain amount of practical significance. However, there are still some questions:

  • The authors did not show how to store soil samples of initial forest soil survey, and the way that stored soil samples may cause changes in soil carbon and nitrogen.
  • The authors referred “the Austrian Forest Soil Survey used the separation of 0-10, 10-20, 20-30, 30-50, 50-80 cm, the BioSoil survey used 0-5, 5-10, 10-20, 20-40, 40-80 cm”, How did the author get the 0-10 cm soil data of the BioSoil survey in Table 1?
  • The statistical analysis part does not show how to obtain the annual change results in Figure 2.
  • The intersection of square meters in all the result figures should be superscript.
  • Why is it not shown in the results the change in soil pH except Table 1?
  • The author did not show the significance level or correlation coefficient of the statistical results in the MS. Is it because all the statistical test results are not significant?

Author Response

pls see attachment

Round 2

Reviewer 1 Report

Second review of the manuscript “Decadal changes of chemical properties of Austrian forest soils”

In relation to the comments from the authors, I maintain that this manuscript should be rejected because it is extremely limited in its scope, methods, and analysis.

I agree that there is always some benefit from presenting data, but some in-depth analysis of such data (whatever data, be it a national survey or any other data set) is to be expected from a manuscript willing to be published in an international journal. And this manuscript fails to do that.

The authors treat all sites as ‘equivalent’, making no attempt to provide any hypothesis about the possible explanatory variables of changes in the studied variables other than the simplistic silicatic/calcareous bedrock dichotomy and the sampling year. The variables studied by the authors may be related to not only these two variables but, among others, soil type, slope aspect, distance to industrial areas, forest type, tree species, vegetation density, forest management, and previous land-use (if different from current) as well. And I would have expected the autors to make some effort to include some of these variables in their analysis.

The distinction between ‘calcareous’ and ‘silicatic’ bedrocks (which, on the other hand, may or may not be the soil parent material) is a very poor one. Even within these two types there are surely quite different types of rocks in Austria, and these differences may be significant for the variables analysed. Barré et al (2017), for example, considered 5 different types of soil parent material to study soil organic carbon (SOC) stocks in a 17 km2 area.

Furthermore, the authors consider that the bedrock being silicatic/calcareous is equivalent to the soils being silicatic/calcareous, which is simply not true. There is also a strong contradiction in the authors stating that “every soil profile (may be the authors mean “every soil horizon”?) where carbonate was detected in the field test was grouped to calcareous soils” (last paragraph of section 2.1) and the data in Table 1 showing that their “calcareous soils” have mean pH values of 6.4, 6.6, or 6.8. Any horizon fizzing when treated with HCl has a pH value certainly over 7.0 if not over 7.5. Nevertheless, I insist that ”silicatic soils” is an unknown soil type, a term that has no practical meaning, and is unacceptable in a scientific paper; and that some horizons in a soil may react to HCl while others may not, and therefore such soil cannot be properly classified in the binomial “silicatic/calcareous” system.

pH has a low sensitivity to change, and other indicators such as acid neutralizing capacity, base saturation, or exchangeable acidity are more appropriate for assessing changes in acid soil horizons. In the case of calcareous horizons, no change in soil pH should have been expected (Blake et al., 1999) as CaCO3 has a very strong buffering capacity, and the determination of changes in calcium carbonate or active lime may have proved more useful.

Depth of sampling is critical to making accurate measurements of changes in SOC stocks and sampling the full soil depth is required to properly account for SOC changes (Van den Bygaart et al., 2011; Chapman et al., 2013). Therefore, measuring SOC to a pre-defined depth, in this case 50 cm, is not adequate to assess possible changes in SOC stocks. Furthermore, apparently all soils have been sampled to a depth of 80 cm, which implies that they all have a depth of at least 80 cm, which is hard to believe.

The method of “slicing the data in 1 cm-steps” seems very odd to me. What does “slicing the data” mean?, to 1 cm?, for what purpose? If you have data for the 20-40 cm and the 40-80 cm depths, the best you can have is an estimate by interpolation but nothing more.

In their new version the authors now estate that “a preliminary unpublished project has shown that the deepest layer of the mineral soil is representative for the entire soil profile [in terms of texture]”. I take this to mean that the authors have considered tha same proportions of sand, silt, and clay for all mineral horizons in a given soil which is certainly stretching things too far.

References

Barré, P. et al. 2017. Geological control of soil organic carbon and nitrogen stocks at the landscape scale. Geoderma, 285: 50-56.

Blake, L., et al. 1999. Changes in soil chemistry accompanying acidification over more than 100 years under woodland and grass at Rothamstead Experimental Station, UK. European Journal of Soil Science, 50: 401-412.

Chapman, S.J.  et al. 2013. Comparison of soil carbon stocks in Scottish soils between 1978 and 2009. European Journal of Soil Science, 64(4): 455-465.

Van den Bygaart, A.J. et al. 2011. Impact of sampling depth on differences in soil carbon stocks in long-term agroecosystem experiments. Soil Science Society of America Journal, 75(1): 226-234.

Reviewer 3 Report

The author answered all the questions and revised the manuscript as required.

Author Response

thank you for the encouraging review